# DoDo-Code: an Efficient Levenshtein Distance Embedding-based Code for 4-ary IDS Channel

**Alan J.X. Guo**[1,2]*, **Sihan Sun**[1], **Xiang Wei**[1], **Mengyi Wei**[1], **Xin Chen**[1,2]

[1]Center for Applied Mathematics, KL-AAGDM, Tianjin University, Tianjin 300072, China
[2]State Key Laboratory of Synthetic Biology, Tianjin University, Tianjin 300072, China
{jiaxiang.guo, sihansun, weixiang, mengyi.wei, chen_xin}@tju.edu.cn

## Abstract

With the emergence of new storage and communication methods, the insertion, deletion, and substitution (IDS) channel has attracted considerable attention. However, many topics on the IDS channel and the associated Levenshtein distance remain open, making the invention of a novel IDS-correcting code a hard task. Furthermore, current studies on single-IDS-correcting code misalign with the requirements of applications which necessitates the correcting of multiple errors. Compromise solutions have involved shortening codewords to reduce the chance of multiple errors. However, the code rates of existing codes are poor at short lengths, diminishing the overall storage density. In this study, a novel method is introduced for designing high-code-rate single-IDS-correcting codewords through deep Levenshtein distance embedding. A deep learning model is utilized to project the sequences into embedding vectors that preserve the Levenshtein distances between the original sequences. This embedding space serves as a proxy for the complex Levenshtein domain, within which algorithms for codeword search and segment correcting is developed. While the concept underpinning this approach is straightforward, it bypasses the mathematical challenges typically encountered in code design. The proposed method results in a code rate that outperforms existing combinatorial solutions, particularly for designing short-length codewords.

## 1 Introduction

With the emergence of new storage and communication methods [1, 2, 3, 4, 5], insertion, deletion, and substitution (IDS) channels over non-binary symbols have attracted significant attention. However, applying existing IDS-correcting codes to practical applications is not straightforward and faces challenges such as a low overall code rate and limited multiple error-correcting capabilities.

Most of the current IDS-correcting codes focus on correcting a single error [6, 7, 8, 9, 10] or a burst of errors [11, 12, 13], with an emphasis on achieving asymptotic optimality in code rate. These codes are typically varieties of the Varshamov-Tenengolts (VT) code [6, 14, 15, 16]. Codebook generation has recently gained attention in specific tasks, such as the $p$-substitution-$k$-deletion code [17].

However, in most applications, the ability to correct multiple IDS errors is crucial. This is because IDS errors may occur simultaneously at different positions along the codeword, and the likelihood of insertion, deletion, and substitution can vary due to channel properties, making them unequal in occurrence [18, 19].

To address the challenge of correcting multiple errors, a compromise approach is to employ segmented error-correcting code, as proposed in previous studies [20, 21, 22, 23, 24, 19]. The sequence is

---

*Corresponding author.

39th Conference on Neural Information Processing Systems (NeurIPS 2025).

implicitly segmented into disjoint segments, each capable of correcting one single IDS error. This segmentation enables the sequence to rectify multiple errors to some extent. However, the possibility remains that multiple errors may occur within the same segment. Researchers can employ an outer error-correcting code (ECC) to address such failures.

Unfortunately, the segmented error-correcting codes usually have low code rate, since these codes use the same code rate as their underlying single-IDS-correcting code within each segment, and current IDS-correcting codes have low code rate for small codeword lengths. For example, an order-optimal code of length $n$ over the 4-ary alphabet using $\log n + O(\log \log n)$ redundancy bits is introduced in [9]. The state-of-the-art code rate till now is proposed in [10] with redundancy bits of $\log n + \log \log n + 7 + o(1)$, which reduces the redundancy by 6 bits compared to the original code in [9]. Although such codes are efficient when $n$ is large, the constant term in the number of redundancy bits limits their code rate when $n$ is small. Given this, there is potential to enhance the code rate of segmented error-correcting codes, because the segments usually use shorter codewords [23].

This work focus on constructing a 4-ary code that uses fewer redundancy bits than the existing order-optimal code offered by the mathematicians [9, 10] at the end of short-length codewords. Namely, following the bounded distance decoder (BDD) which is one of the basics of classical codes [25], the novel DoDo-Code is proposed by leveraging the deep embedding of the Levenshtein distance. The embedding space is utilized as a proxy for the intricate Levenshtein domain, facilitating high code rate codebook design and fast IDS-correcting. The proposed DoDo-Code uses a comprehensive approach that includes the following key procedures: deep embedding of the Levenshtein domain, deep embedding-based greedy search of codewords, and deep embedding-based segment correcting.

As a result, the proposed DoDo-Code offers a solution for IDS-correcting code when code length is short, excelling in the following aspects:

- The proposed DoDo-Code achieves a code rate that surpasses the state-of-the-art and shows characteristics of "optimality" when code length $n$ is small.
- With one edit operation corrupted codewords can be firmly corrected, the computational complexity is effectively reduced to $O(n)$ of the decoder to correct IDS errors.

To the best of our knowledge, the proposed DoDo-Code is the first IDS-correcting code designed by deep learning methods and the first IDS-correcting code that shows characteristics of "optimality" when the code length is small.

## 2 Related works

While neural channel codes [26, 27, 28, 29, 30, 31] have recently gained attention and achieved state-of-the-art performance in several settings, they offer little help in addressing the aforementioned issue. This is because they primarily target the additive white Gaussian noise (AWGN) channel and handle flip or erasure errors, rather than IDS errors. Moreover, being neural network-based, these solutions typically fail to generate the static, explicit codebooks for downstream applications.

## 3 Bounded Distance Decoder

The proposed code employs a quite fundamental approach called the BDD; let's revisit the basics of classical codes [25].

Given a code $C$, which is a collection of codewords, its minimum distance is defined as follows:

$$d(C) = \min\{d(\boldsymbol{c}_i, \boldsymbol{c}_j) : \boldsymbol{c}_i, \boldsymbol{c}_j \in C, i \neq j\}. \tag{1}$$

Once a code $C$ with a minimum distance of $d$ is constructed, a BDD can be deployed with a decoding radius $r = \lfloor \frac{d-1}{2} \rfloor$ by correcting a corrupted word $\hat{\boldsymbol{c}}$ to the corresponding codeword $\boldsymbol{c}$ such that $d(\hat{\boldsymbol{c}}, \boldsymbol{c}) \leq r$.

In the context of correcting IDS errors, the Levenshtein distance [7] plays a pivotal role. It is defined as the minimum number of insertions, deletions, and substitutions required to transform one sequence into another. According to the principles of BDD, to correct a single IDS error, the decoding radius

in terms of Levenshtein distance is $r = \lfloor \frac{d-1}{2} \rfloor = 1$, meaning the minimum distance of the code must be at least $d = 3$. Therefore, constructing a code $C(n)$ for a fixed code length $n$, whose elements have mutual Levenshtein distances greater than 3, is considered.

However, the construction of such a code faces three significant challenges. Firstly, the sizes of Levenshtein balls, representing the set of a sequence and its neighboring sequences within a Levenshtein distance of $r$, exhibit a lack of homogeneity [32, 33]. Researchers want to select sequences with small Levenshtein balls as the codewords to enhance the code rate, but a clear depiction of Levenshtein balls is still absent [32, 33]. An algorithm surpassing random codeword selecting has not yet been published to the best of our knowledge. Secondly, the computational complexity of the Levenshtein distance-based BDD is substantial. The complexity of computing the Levenshtein distance is at least $O(n^{2-\epsilon}), \forall \epsilon > 0$ [34, 35], unless the strong exponential time hypothesis is false. Additionally, most existing neighbor searching algorithms, which are keys to the BDD process, are primarily designed for conventional distance [36] and inapplicable to the Levenshtein distance.

## 4  DoDo-Code

### 4.1  Deep embedding of Levenshtein distance

Considering the complexity of calculating the Levenshtein distance, researchers have explored mapping sequences into embedding vectors, using a conventional distance between these vectors to approximate the Levenshtein distance [37, 38]. Recently, deep learning techniques have been employed for Levenshtein distance embedding and achieved remarkable performance across various works [39, 40, 41, 42, 43, 44].

From a broader perspective, it is found that these embeddings not only accelerate the Levenshtein distance estimation, but also offer a way to analyze the properties and structures of the Levenshtein distance. The embedding aims to create a vector space where the squared Euclidean distance between vectors serves as a proxy for the Levenshtein distance between the original sequences. This allows us to leverage the geometric structure of the embedding space to reason about the complex combinatorial properties of the Levenshtein domain. For example, in the vector space, the embedding vectors of sequences within a Levenshtein ball should naturally exhibit a tight clustering. The embedding model from [44] is modified to focus more on the Levenshtein neighbor relations between the sequences in our work.

Let $s$ and $t$ denote two sequences of length $n$ on the alphabet $\{0, 1, 2, 3\}$, and let $d = \Delta_L(s, t)$ represent the groundtruth Levenshtein distance between them. Our task is to identify an embedding function $f(\cdot)$, such that the mapped embedding vectors $u = f(s)$ and $v = f(t)$ have a conventional distance $\hat{d} = \Delta(u, v)$ approximates to the groundtruth Levenshtein distance $d = \Delta_L(s, t)$. Let the embedding function $f(\cdot; \theta)$ be a deep embedding model with learnable parameters $\theta$, which is implemented as a model with 10 1D-CNN [45] layers and one final batch normalization [46] in this study. The training of $f(\cdot; \theta)$ can be expressed as an optimization of:

$$\hat{\theta} = \underset{\theta}{\arg\min} \sum \mathcal{L}(d, \hat{d}; \theta) \tag{2}$$

$$= \underset{\theta}{\arg\min} \sum \mathcal{L}(d, \Delta(f(s; \theta), f(t; \theta))), \tag{3}$$

where the function $\mathcal{L}(\cdot, \cdot)$ is a predefined loss function that measures the disparity between the predicted distance and the groundtruth distance. By optimizing (2), the parameters of the embedding model are learned and denoted as $\hat{\theta}$, and the optimized deep embedding model $f(\cdot; \hat{\theta})$ is capable of mapping sequences to their corresponding embedding vectors. This model configuration is often referred to as a Siamese neural network [47]. A brief illustration of the utilized Siamese neural network is presented in Figure 1.

The squared Euclidean distance between the embedding vectors is employed as the approximation of the groundtruth Levenshtein distance. It is defined as:

$$\hat{d} = \Delta(u, v) = \sum_i (u_i - v_i)^2. \tag{4}$$

Although the squared Euclidean distance is not a true distance metric, its effectiveness has been validated in [43]. Moreover, since it is simply the square of the Euclidean distance, it remains compatible with most neighbor searching algorithms in Euclidean space.

The negative log-likelihood loss with the Poisson distribution (PNLL) [44], which is formulated as:

$$\mathcal{L}(d, \hat{d}) = \text{PNLL}(\hat{d}; d) = \hat{d} - d \ln \hat{d}, \quad (5)$$

has been proposed to provide a global approximation of the Levenshtein distance. In this work, the code's construction is dependent on local structures within the Levenshtein distance domain, namely sequences at a distance of 1 or 2. Consequently, from the perspective of the greedy search algorithm, there is no functional difference between using the complete Levenshtein distance and a truncated version. However, this distinction is raised for training the embedding model, relaxing the optimization ob-

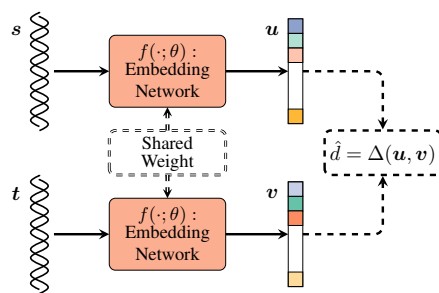

Figure 1: Siamese neural network. Given two sequences $s, t$, mapped to respective embedding vectors $u, v$. The approximated distance is calculated as a conventional distance between $u$ and $v$.

jective to a truncated Levenshtein distance is an easier learning task for the model than approximating the global Levenshtein distance metric precisely. In view of this, the loss function is revised to emphasize the approximations between sequence pairs within the Levenshtein balls of radius 2. Specifically, the model is trained to provide a precise prediction for Levenshtein distance 1 and to ensure that the predicted distance is greater than 2 when the groundtruth distance is greater or equal to 2. The revised loss function is defined as follows:

$$\tilde{\mathcal{L}}(d, \hat{d}) = \begin{cases} \mathcal{L}(d, \hat{d}) & \text{if } d = 1; \\ \mathbf{1}_{\hat{d} < 2} \cdot \mathcal{L}(2, \hat{d}) & \text{if } d \geq 2, \end{cases} \quad (6)$$

where $\mathbf{1}_{\hat{d} < 2}$ is the indicator function that evaluates to 1 when $\hat{d} < 2$.

For the sake of brevity, $f(\cdot)$ will be used to represent the learned embedding function $f(\cdot; \hat{\theta})$ in the subsequent discussion.

## 4.2 Deep embedding-based greedy search of codewords

As previously mentioned, single-IDS-correcting codes with large code lengths $n$ are ineffective, as the longer the codeword, the higher the likelihood of multiple errors occurring within the same segment. Moreover, existing combinatorial codes have already achieved order optimal code rates, which are nearly optimal when $n$ is large.

In view of this, the single-IDS-correcting codes with small code lengths $n$ and aim to achieve higher code rates are focused. By concentrating on smaller code lengths, the random search for a codebook becomes feasible.

A random search-based approach of constructing the code $C(n)$ is repeating the following procedure: randomly selecting a sequence from a candidate set (initially consists of all possible sequences $A(n) = \{0, 1, 2, 3\}^n$), and then filtering out the neighboring sequences of the chosen one from this set.

To outperform the random codeword selecting algorithm in terms of code rate, a selecting criterion for choosing codewords from the candidate set is crucial in the greedy search procedure. Finding a method to select more codewords is equivalent to enhancing the overall code rate. In a greedy search approach, codewords with fewer neighbors should be selected in advance. However, it remains an open problem to accurately depict the neighbor density of each sequence, as only the minimum, maximum, and average sizes of Levenshtein balls with radius 1 have been studied in existing works [32, 33].

Fortunately, the deep embedding of Levenshtein distance establishes a connection between the structural characteristics of Levenshtein distances and the distribution properties of the embedding vectors. This deep embedding allows for a rough estimation of the neighboring sequences associated

with a given sequence $\boldsymbol{s}$ through the Euclidean ball centered at the embedding vector $\boldsymbol{u} = f(\boldsymbol{s})$. By employing a final batch normalization, the embedding model outputs vectors that follow a multivariate normal distribution $N(\boldsymbol{0}, \boldsymbol{\Sigma})$ with a mean vector $\boldsymbol{0}$ and a covariance matrix $\boldsymbol{\Sigma}$. This probability density function (PDF) of the embedding vectors can then be leveraged to evaluate the density of neighbors around the codewords. Namely, low-density vectors correspond to sequences that have fewer Levenshtein neighbors. By selecting these sequences first, the greedy search makes the efficient choice at each step, leaving maximal room for future codewords and thus maximizing the final codebook size.

Let $m$ denote the dimension of the random vector, the PDF of $N(\boldsymbol{0}, \boldsymbol{\Sigma})$ is formulated as

$$p(\boldsymbol{x}) = (2\pi)^{-\frac{m}{2}} |\boldsymbol{\Sigma}|^{-\frac{1}{2}} \exp\left(-\frac{1}{2}\boldsymbol{x}^T\boldsymbol{\Sigma}^{-1}\boldsymbol{x}\right). \tag{7}$$

The covariance matrix $\boldsymbol{\Sigma}$ is easy to estimate with all of the embedding vectors $f(A(n))$. Having the estimated covariance matrix $\hat{\boldsymbol{\Sigma}}$ in hand, the PDF of $N(\boldsymbol{0}, \hat{\boldsymbol{\Sigma}})$ for each embedding vector can be calculated using (7). To achieve the goal of selecting codewords with fewer neighbors, an effective selecting criterion can be expressed in the embedding space as the sequence whose embedding vector has the lowest PDF value should be chosen as the codeword in each iteration. A step further, by making some simplifications to (7), an embedding vector $\boldsymbol{u}$ and its corresponding sequence should be selected if $\boldsymbol{u}^T\hat{\boldsymbol{\Sigma}}^{-1}\boldsymbol{u}$ is the maximum over the candidate set.

---

**Algorithm 1** Deep embedding-based greedy search of codewords

---

**Input:** Codeword length $n$; the 4-ary alphabet $\Sigma_4$; a pre-trained embedding model $f(\cdot)$.
**Output:** A codebook $C(n)$ where the minimum distance between any two codewords is at least 3.
 1: Create the candidate set $A \leftarrow A(n)$ containing all $4^n$ sequences of length $n$.
 2: Initialize an empty codebook $C \leftarrow \emptyset$.
 3: Compute the embedding vectors: $U = \{f(\boldsymbol{s})|\boldsymbol{s} \in A(n)\}$.
 4: Estimate the covariance matrix $\hat{\boldsymbol{\Sigma}}$ of the embedding vectors $U$.
 5: **while** $A \neq \emptyset$ **do**
 6: $\quad$ Select $\boldsymbol{s}$ from $A$ that $f(\boldsymbol{s})$ have the lowest PDF value: $s = \arg\max_{\boldsymbol{s} \in A} f(\boldsymbol{s})^T\hat{\boldsymbol{\Sigma}}^{-1}f(\boldsymbol{s})$.
 7: $\quad$ Add $\boldsymbol{s}$ to the codebook: $C \leftarrow C \cup \{\boldsymbol{s}\}$.
 8: $\quad$ Remove neighboring sequences: $A \leftarrow A \backslash B(\boldsymbol{s}, 2)$, where $B(\boldsymbol{s}, 2) = \{\boldsymbol{s}' \in A|\Delta_L(\boldsymbol{s}, \boldsymbol{s}') \leq 2\}$.
 9: **end while**
10: **return** $C(n) = C$.

---

The entire procedure for the greedy selecting of codewords is illustrated in Algorithm 1. Firstly, the statistical distribution $N(\boldsymbol{0}, \hat{\boldsymbol{\Sigma}})$ is estimated on the embedding vectors $f(A(n))$. Subsequently, the sequence from the candidate set whose embedding vector possesses the lowest PDF value is chosen as a codeword. Finally, the candidate set is updated by filtering out the Levenshtein ball, and this selecting iteration is repeated until the candidate set becomes empty.

Once the codebook is generated, users can encode information by choosing codewords according to indices from a predefined order.

### 4.3 Deep embedding-based segment correcting

Refer a corrupted codeword as a segment. In the Levenshtein domain, only brute-force methods are applicable for segment correcting to the best of our knowledge. This segment correcting method in BDD involves calculating the Levenshtein distance between the segment and all the codewords, and then selecting the codeword with the minimal distance as the correction. However, this brute-force approach incurs significant computational complexity, scaling up to $O(n^2|C(n)|)$. It is worth noting that the cardinality of the code $|C(n)|$ grows fast with an increasing $n$ in the experiments. While this method can undoubtedly be optimized through techniques like early stopping when calculating Levenshtein distances, its complexity remains at least $O(|C(n)|)$.

In this work, the deep embedding vectors and their distances are leveraged to correct errors without using the Levenshtein domain. The segment correcting procedure is outlined in Figure 2. To be precise, a K-dimension tree (K-d tree) [48] is constructed from the embedding vectors $f(C(n))$

corresponding to the codewords $C(n)$. Subsequently, the embedding vector $\hat{v} = f(\hat{c})$ of a corrupted segment $\hat{c}$ is utilized to query its nearest neighbors $v = f(c)$ within this K-d tree, thereby confirming the nearest codeword $c$ to this segment. Significantly, the construction of the K-d tree incurs a one-time complexity cost, while the average complexity of querying operations from the K-d tree is $O(\log |C(n)|)$ [49], representing a considerable improvement over the previously mentioned brute-force search method.

It is important to recognize that this neighbor-searching procedure is conducted within the embedding space, and is based on an approximation rather than the exact Levenshtein distance. As a result, the query results from the K-d tree may not always accurately represent the true minimum Levenshtein distances. To mitigate this, multiple nearest neighbors can be queried, and the Levenshtein distances between the segment and the queried neighboring codewords can be double-checked to improve the reliability of the results.

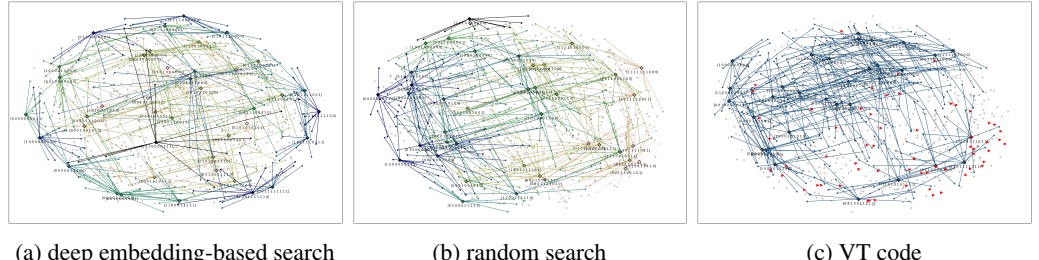

Figure 2: Flowchart for the deep embedding-based segment correcting. A K-d tree is constructed using the embedding vectors $f(C(n))$ of all the codewords. The embedding vector of a segment is used to query the K-d tree for its neighboring sequences.

## 5 Experiments and Results

### 5.1 Codewords in the embedding space

To demonstrate the effectiveness of the proposed deep embedding-based greedy search of codewords and the reasonability of using an estimated distribution to evaluate the density of the sequences, the embedding vectors for all the candidate sequences and the selected codewords are visualized and presented in Figure 3.

| (a) deep embedding-based search | (b) random search | (c) VT code |

Figure 3: The relationships between the codewords and their neighboring sequences in the embedding space under three different scenarios: (a) results from the proposed deep embedding-based codeword search; (b) results from a random codeword search; (c) results using codewords from the VT code. In each subplot, the diamond markers represent the codewords, and the solid lines connect the codewords to their distance 1 neighbors. In (a) and (b), the color indicates the order in which the codewords were selected, with darker colors signifying codeword selected earlier. In (c), the red triangle markers identify sequences that are neither codewords nor within a distance of 2 to any codeword.

To enhance the readability, Figure 3 is generated from experiments with a simplified setting: the codeword length is set to $N = 10$, the code is reduced from a 4-ary alphabet to a binary alphabet, and the embedding dimension is reduced to 8. To visualize the embedding vectors effectively, the $t$-SNE [50] is employed to project the high-dimensional embedding vectors into $\mathbb{R}^2$. The Figure 3a, Figure 3b, and Figure 3c are plotted by the codewords/vectors obtained from the proposed deep embedding-based codeword search, a random codeword search, and the VT code, respectively. In each subfigure, the diamond markers denote the codewords/vectors, and the solid lines connect these to their distance-1 neighbors.

In Figure 3a and Figure 3b, the color scheme indicates the order in which the codewords were selected, with darker colors representing codewords selected earlier in the search procedure. A comparison of these two subfigures, which represent the projections of the embedding vectors, shows that the deep embedding-based search tends to select codewords closer to the periphery of the estimated distribution

of embedding vectors, while the random search selects codewords without any specific pattern. This observation aligns with that the embedding vectors follow a multivariate normal distribution, and the vectors that deviate from the mean should be selected earlier in Figure 3a due to their lower PDF in the estimated distribution.

In Figure 3c, the combinatorially constructed codewords from the VT code are plotted over the embedding vectors. The red triangle markers identify the isolated sequences that are neither codewords nor within a distance of 2 from any codeword, which isolated sequences would be eliminated during the greedy codeword search in the first two subfigures. The presence of these isolated sequences suggests that the Levenshtein balls with a radius 2, centered on the VT codewords, cannot make a complete coverage on the Levenshtein domain. This indicates that a modification of the VT code, which considers these isolated sequences, could achieve a larger code rate. Additionally, it is also observed in Figure 3c that the VT codewords have a biased distribution in the embedding space used in this experiment, with the codewords tending to be located in the North-West, while the isolated sequences are more likely found in the South-East of the plane.

## 5.2 Code rate and optimality

To illustrate the utilization of the deep embedding-based greedy search yields an augmented count of codewords, thereby promoting the overall code rate. Comparisons are made between the proposed code and the state-of-the-art combinatorial codes.

Given that the codebook, once generated, is relatively independent of the deep learning model, the code with the maximum cardinality from among 10 runs of the computational experiments is selected. The corresponding code rates are calculated using the formula

$$r(n) = \frac{\log_4 |C(n)|}{n}. \tag{8}$$

### 5.2.1 DoDo-Code outperforms the combinatorial codes.

The resulting code rates are visualized against the code length in Figure 4. For comparison, the code rates of the combinatorial code introduced in [9] which is order-optimal with redundancy of $\log n + O(\log \log n)$ bits, as well as the state-of-the-art code rates from [10] which is also order-optimal using $\log n + \log \log n + 7 + o(1)$ redundancy bits, are presented. Additionally, the code rates corresponding to the imaginary redundancies of $\log n + \log \log n + 1$, $\log n + \log \log n + \log 3$, and $\log n + \log \log n + 2$ in Figure 4 are also drawn. It's worth noting that for small code lengths $n$, no existing 4-ary code achieves these levels of redundancies.

Focusing solely on the proposed code, Figure 4 clearly shows a trend of increasing code rates with longer codeword lengths $n$. Notably, for $n = 11$, the code rate reaches $68.9\%$. Compared to existing state-of-the-art works [9, 10], the DoDo-Code achieves a significantly higher code rate. This improvement is attributed to the fact that, although these established combinatorial codes are order optimal, the constant terms in their redundancies dominate when $n$ is small, thereby reducing the code rates.

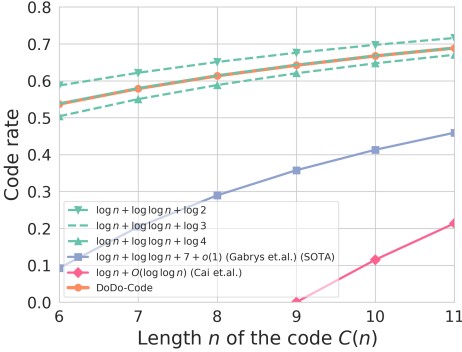

### 5.2.2 DoDo-Code may represent the minimal redundancy achievable.

Furthermore, when compared to the imaginary redundancies of $\log n + \log \log n + \{\log 2, \log 3, \log 4\}$ the code rate curve of DoDo-Code lies between $\log n + \log \log n + 1$ and $\log n + \log \log n + 2$ and overlaps the curve of $\log n + \log \log n + \log 3$ exactly. It may be claimed that the proposed DoDo-Code is ap-

Figure 4: The code rate of code searched by deep embedding-based greedy search strategy, reported as the best in 10 runs. Comparison methods are the state-of-the-art approaches of Cai and Garbys. The code rates corresponding to order-optimal redundancies of $\log n + \log \log n + 1$, $\log n + \log \log n + \log 3$, and $\log n + \log \log n + 2$ are also plotted. These levels of redundancies are not achieved by any codes before this work.

proximately optimal using $\log n + \log \log n + \log 3$ redundancy bits. However, this assertion lacks theoretical proof in this study.

It's worth noting that $\log n + \log \log n + \log 3$ can be reformulated as $\log 3n + \log \log n$, which is nearly the best code rate for a single-IDS-correcting code. It is known that correcting a one-bit flip in $N$ bits requires at least $\log N$ redundancy bits for information to identify the error position. In the context of a 4-ary code, correcting a substitution in $n$ bases requires information on both the error position and the substituted letter, which could be any of the other three letters. As a result, a minimum of $\log 3n$ redundancy bits is required. Given this, the redundancy $\log 3n + \log \log n$ is very close to its lower bound and possibly represents the minimal redundancy achievable. Mathematicians might explore this topic further by leveraging combinatorial or probabilistic methods.

## 5.3   Ablation study on embedding space searching and revised PNLL loss

To illustrate the effectiveness of searching for codewords in the embedding space, the comparison is made between the proposed embedding space search and the random codeword selecting, which is also introduced for the first time to the best of our knowledge. The cardinalities of the codebooks were compared, both in terms of average and maximum results across 10 runs, with the findings presented in Table 1. The results clearly indicate that using the PDF of the distribution of embedding vectors as the selecting criterion in the greedy search yields larger codebooks. Furthermore, it is observed that the increase in codeword count becomes more pronounced as the codeword length $n$ increases. For instance, when $n = 11$, the deep embedding-based greedy search identifies $16.8\%$ more codewords compared to the random codeword selecting approach.

For an ablation study on using the revised PNLL loss in Equation (6) as the optimization target for the Levenshtein distance embedding network, experiments were also conducted engaging the original PNLL loss from Equation (5), with the results presented in the row labeled DEGS* in Table 1. The results suggest that employing the revised PNLL loss slightly increased the number of searched codewords.

Table 1: The cardinality of constructed codebook. The results are reported as the mean value and maximum value over 10 runs of the experiments. The method "Rand" stands for random codeword selecting method, the method "DEGS" (resp. "DEGS*") stands for the proposed deep embedding-based greedy search with the revised PNLL loss (resp. original PNLL loss), and the "$\Delta$" stands for the differences.

|  | Method | $n = 7$ | $n = 8$ | $n = 9$ | $n = 10$ | $n = 11$ |
|---|---|---|---|---|---|---|
| avg. | Rand | $251.5 \pm 5.1$ | $813.2 \pm 3.7$ | $2694.0 \pm 15.2$ | $9091.7 \pm 18.8$ | $31071.9 \pm 40.5$ |
|  | DEGS* | $264.5 \pm 3.3$ | $873.3 \pm 8.5$ | $2963.5 \pm 18.4$ | $10199.4 \pm 57.6$ | $35720.4 \pm 297.7$ |
|  | DEGS | $267.5 \pm 4.7$ | $884.8 \pm 15.3$ | $3001.6 \pm 8.5$ | $10325.4 \pm 48.6$ | $35973.1 \pm 157.8$ |
|  | $\Delta$ | $+6.4\%$ | $+8.8\%$ | $+11.4\%$ | $+13.6\%$ | $+15.8\%$ |
| max. | Rand | 259 | 820 | 2717 | 9124 | 31142 |
|  | DEGS* | 270 | 887 | 2983 | 10283 | 36191 |
|  | DEGS | 275 | 900 | 3011 | 10414 | 36368 |
|  | $\Delta$ | $+6.2\%$ | $+9.8\%$ | $+10.8\%$ | $+14.1\%$ | $+16.8\%$ |

## 5.4   Success rate and experimental time complexity of segment correcting

The proposed deep embedding-based segment correcting is proposed as an alternative to the neighboring search procedure of BDD. Experiments were conducted to demonstrate that the proposed method is both reliable and efficient.

It is worth noting that the searched codewords maintain a minimum mutual Levenshtein distance of 3, ensuring that a single error in a codeword can be confidently corrected by the BDD. However, the deep embedding of the Levenshtein distance introduces approximation error, which can affect the reliability of the nearest neighbors identified through the tree search in the embedding space. A compromise solution is to increase the number $k$ of searched neighbors, and then perform a double confirmation using the Levenshtein distances to these $k$ neighbors. Experiments with different values of $k$ were conduceted, and the number of failed corrections out of $10^8$ attempts is presented in Table 2.

As indicated in Table 2, the ratio of failed corrections ranges from $0.3\%$ to $0.9\%$ along with different code length, when only one embedding vector ($k = 1$) is queried. When the search is expanded to $k = 2$ neighbors, the number of failed corrections decreases significantly. Further increasing the searched neighbors to $k = 4$, the number of failed corrections is 0 out of correcting $10^8$ modified codewords.

Table 2: Number of failed segment correctings in $10^8$ tries by using tree search. The segments are obtained by randomly one edit modification on the codewords. $k$ is the number of neighbors queried in the tree search.

|        | $n = 7$ | $n = 8$ | $n = 9$ | $n = 10$ | $n = 11$ |
|--------|---------|---------|---------|----------|----------|
| $k = 1$ | 328,142 | 398,439 | 465,080 | 740,468  | 828,885  |
| $k = 2$ | 0       | 4,411   | 2,330   | 7,060    | 10,869   |
| $k = 3$ | 0       | 0       | 754     | 0        | 121      |
| $k = 4$ | 0       | 0       | 0       | 0        | 0        |
| $k = 5$ | 0       | 0       | 0       | 0        | 0        |

The proposed deep embedding-based segment correcting utilizes a K-d tree search in Euclidean space, which theoretically offers a lower average complexity of $O(\log |C(n)|)$. To demonstrate the efficiency of this method, experiments varying the number $k$ of searched neighbors were performed, compared with the brute-force search method, which corrects segments by identifying the codeword with the minimal Levenshtein distance. As shown in Figure 5, the proposed method in Euclidean space significantly reduces time complexity by orders of magnitude compared to the brute-force approach. Moreover, the extra burden of raising $k$ from 1 to 5 is minimal, as indicated in Figure 5.

## 5.5 Complexity

When the complexity of copying a codeword is disregarded, the encoder of the DoDo-Code operates with negligible complexity, since the codebook is pre-generated, and encoding simply consists of selecting a codeword by its index.

The embedding model, which is implemented by a CNN architecture, maps the sequences to their embedding vectors with a complexity of $O(n)$. Without considering the one-time cost of building the K-d tree by the embedding vectors of the codebook, the segment correcting process incurs a time complexity of at most $O(n)$ when querying $k = 1$ neighboring sequence. The set $A(n)$ containing all sequences of length $n$ over the 4-ary alphabet has a cardinality $4^n$, and the code $C(n)$ is a subset of $A(n)$. The theoretical expected query time for the K-d tree is $O(\log |C(n)|)$, which simplifies to $O(n)$, considering $|C(n)| < 4^n$. When querying $k > 1$ neighbors, the double-check on Levenshtein distance increases the time complexity to $O(n^2)$.

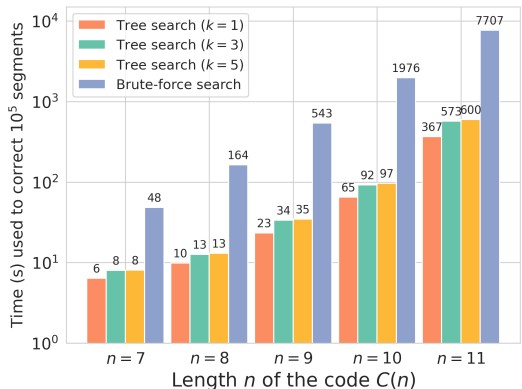

Figure 5: The time used to correct $10^5$ segments. The segments are obtained by randomly one edit modification on the codewords. $k$ is the number of neighbors in the tree search. The brute-force search calculates the Levenshtein distances until the finding of a 1-distance codeword. The $y$-axis is in $\log$ scale.

The memory complexity to store the K-d tree is $O(m|C(n)|)$, where $m$ is the dimension of the embedding vectors and $|C(n)|$ is the cardinality of the codebook. Since the tree can be generated on-the-fly from the codebook each time the decoder is initialized, and the codebook is deterministically generated using a given embedding model and random seed, the only persistent storage needed is for the embedding model itself.

## 5.6 Dataset, source code, and model setting

All the sequences used for training and testing are generated randomly. The groundtruth Levenshtein distance is obtained by a Python module called Levenshtein. Therefore, the experiments run inde-

pendently of any specific dataset and generate the data on their own. The source code is available in `https://github.com/aalennku/DoDo-Code`. Unless otherwise specified, the embedding model utilizes an architecture of stacking 10 1D-CNNs, with the embedding vector dimension set to $64$.

## 6 Conclusion and Limitations

To address the code rate issue from the segmented error-correcting codes, the DoDo-Code, which boasts an "optimal" code rate at the short code lengths for $4$-ary IDS correcting code, was proposed. By leveraging the deep embedding space as a proxy for the complex Levenshtein domain, the mathematically unexplored field is bypassed in the code design. The fundamental concept of BDD forms the backbone of the proposed code. In the embedding space, an efficient codeword searching algorithm was introduced to maximize the codebook. Later, the decoding or correcting algorithm was integrated into the Euclidean embedding space by a K-d tree, reducing the computational complexity. Experiments illustrated the proposed DoDo-Code outperforms the state-of-the-art combinatorial codes in code rate when the code length is small.

**Limitations.** The DoDo-Code did not provide explicit mathematical rules for description, necessitating a deeper understanding of combinatorial underlying principles. The codeword searching relies on a greedy search strategy, which can lead to significant complexity when attempting to construct the codebook for large code lengths. We hope that future research by mathematicians will uncover the underlying principles governing codewords and lead to the invention of mathematically defined codes.

## Acknowledgement

This work was supported by the National Key Research and Development Program of China under Grant 2020YFA0712100 and 2025YFC3409900, the National Natural Science Foundation of China, and the Emerging Frontiers Cultivation Program of Tianjin University Interdisciplinary Center.

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
