# OpenReview forum: "DoDo-Code: an Efficient Levenshtein Distance Embedding-based Code for 4-ary IDS Channel"
_NeurIPS.cc/2025/Conference — NeurIPS 2025 poster_

### Official Review · Reviewer_1PZb · 2025-07-02

**Clarity:** 4
**Significance:** 3
**Originality:** 3
**Rating:** 5
**Confidence:** 3

**Summary:**

The paper introduces a novel algorithm for designing high-code-rate single-IDS-correcting codewords through a deep Levenshtein distance embedding. The method is composed from a CNN based Siamese neural network that embeds symbol sequences in a Euclidian sub-space. The embedding is trained to preserve the Levenshtein distance between the original sequences.

The learned embeddings offer a solution for IDS channels decoding in two main aspects:

1. The proposed DoDo-Code achieves a code rate that surpasses the state-of-the-art and shows characteristics of “optimality” when code length n is small.

2. With one edit operation corrupted codewords can be firmly corrected, the computational complexity is effectively reduced to O(n) of the decoder to correct IDS errors.

**Questions:**

1. At what code length n does the DoDo-Code become computationally impractical? Have you conducted experiments for larger values of n, and does the performance advantage over existing methods persist at those scales?

**Ethical Concerns:**

["NO or VERY MINOR ethics concerns only"]

**Final Justification:**

The paper proposes an original and interesting method for code-book design and decoding on single IDS channels. While the approach entails non-trivial space overhead that could limit its immediate practicality, the underlying ideas are innovative and, in my view, warrant publication.

**Limitations:**

The overhead in terms of space is not addressed.

**Paper Formatting Concerns:**

No Formatting issues.

**Quality:**

3

**Strengths And Weaknesses:**

Strengths:

1. The paper is well written and easy to understand.
2. The proposed DoDo-Code achieves a state-of-the-art code rate compared to existing methods in the literature.
2. The learned Levenshtein distance embedding enables a K-d tree–based segment correction procedure. This procedure has a time complexity between O(n) and O(n^2), depending on the number of queries, offering a performance advantage over existing methods.
3. The reported experiments validate the code rate and correction time complexity claims for small code length (n).

Weaknesses:

1. From a ML and deep learning perspective, the work presents no methodological advancements.
2. The empirical evaluation of the correction capabilities and time complexity of the DoDo-Code are very limited, and is applied only on segments obtained by randomly one edit modification on the codewords. No evaluations on real world data sequences/segments were provided.
3. Space complexity is not addressed, as the proposed method require storing the K-d tree and trained NNs together with the DoDo-Code code book.
4. The use of the DoDo-Code for larger values of n is not discussed.
5. The claim for "minimal redundancy achievable" is not formally proved.

---

> ### Author Rebuttal · Authors · 2025-07-29
>
> We sincerely thank the reviewer for their thorough assessment and encouraging feedback. We are encouraged the reviewer  recognized that DoDo-Code achieves state-of-the-art performance with a novel, efficient decoding procedure.
>
> We will address the weaknesses and questions in the follow.
>
> -----
> ## On Methodological Advancement in Machine Learning
>
> While we agree that the underlying network architecture is a well-established deep learning model, this work still presents a novel and methodologically application of deep learning to solve a challenging problem in coding theory. The proposed work demonstrates a new, learning-based heuristic for combinatorial search.
>
> + **A deep embedding-based greedy search**: We are the first to propose a data-driven greedy search that uses the probability density function (PDF) of embedding vectors to select optimal codewords. This approach provides a principled way to maximize codebook size by selecting sequences with fewer neighbors first. As shown in our ablation study, this search method finds up to 16.8% more codewords than a random search, demonstrating its effectiveness.
>
> + **A revised loss function distinct from standard application**: The loss function is tailored to the specific requirements of the BDD. The revised PNLL loss function (Equation 6) specifically focuses on creating a precise mapping for Levenshtein distances of 1 and 2, which is critical for single-error correction, rather than a general-purpose approximation.
>
> + We'd also like to highlight that the NeurIPS 2025 **Call For Papers**, explicitly welcomes submissions in areas of "**Applications**" and "**Machine learning for sciences**" and emphasizes that “The NeurIPS 2025 is **an interdisciplinary conference**”. We believe our work, which introduces a new, learning-based heuristic for combinatorial search, fits squarely within this scope.
>
> -----
> ## On the Scope of Empirical Evaluation
>
> We acknowledge that the experiments were conducted on segments generated by random single-edit modifications. This was a deliberate choice to establish the fundamental performance of DoDo-Code and enable a fair comparison with existing state-of-the-art combinatorial codes and theoretical bounds. These theoretical methods are themselves designed to correct a single error, irrespective of the channel's specific error characteristics. This serves as the standard way to validate the code's fundamental capabilities.
>
> Regarding the correction of single IDS errors with specific real-world patterns, we believe that the experiments partly provide substantial coverage. The experiments detailed in Table 2 tested the code against $10^8$ random single-edit modifications, a scale sufficient to encompass a vast range of potential error cases.
>
> We acknowledge that correcting multiple errors from real-world data involves more complex error patterns. The DoDo-Code proposed in this work focuses on correcting a single error and, therefore, cannot be directly applied to multiple errors. However, it is well-suited to function as a “in-segment” code for schemes like segmented or marker codes [1-4], which are designed to handle multiple errors by correcting one error per segment. In this context, the performance on real-world multiple-error patterns is more related to the design of the segment code architecture, rather than “in-segment” code.
>
> + [1] ZM Liu and M Mitzenmacher. Codes for deletion and insertion channels with segmented errors. IEEE Transactions on Information Theory
> + [2] M Abroshan, R Venkataramanan, and AG Fàbregas. Coding for segmented edit channels. IEEE Transactions on Information Theory
> + [3] K Cai, HM Kiah, M Motani, and TT Nguyen. Coding for segmented edits with local weight constraints. ISIT
> + [4] ZH Yan, C Liang, and HM Wu. A segmented-edit error-correcting code with resynchronization function for DNA-based storage systems. IEEE Transactions on Emerging Topics in Computing
>
> We will consider the application of this work to the correction of multiple errors using real-world data as a future investigation topic.
>
> -----
> ## On Space Complexity
> We thank the reviewer for pointing out the omission of a detailed space complexity analysis. The storage requirements of our method are manageable and consist of two main components:
>
> **The Trained Neural Network**: The embedding model is a 10-layer 1D-CNN with a fixed number of parameters. For a codeword length of $n=10$, the total number of parameters in the network is **2.7m**, making it a computationally lightweight model compared to contemporary deep learning architectures.
>
> **The K-d Tree**: The K-d tree is constructed from the embedding vectors of the final codebook, $C(n)$. Its space complexity is $O(m|C(n)|)$, where $m$ is the embedding dimension (64 in our work) and $|C(n)|$ is the number of codewords. Since codebook generation is a one-time, offline process, and our work focuses on small $n$, this storage overhead is practical and fixed before any decoding occurs. Furthermore, it is worth noting that this storage is not mandatory; the K-d tree can be generated on-the-fly from the codebook each time the decoder is initialized, thus minimizing persistent storage requirements.
>
> For the online decoding process, the user only needs the lightweight embedding model and the pre-built K-d tree, not the entire training or search apparatus.
>
> -----
> ## On Scalability for Larger Codeword Length $n$
> As noted in the paper's limitations section, the greedy search for codebook generation becomes computationally intensive for large $n$ due to the need to process all $4^n$ possible sequences.
>
> This is a deliberate trade-off. Our work is specifically motivated by the poor performance of existing order-optimal codes at short lengths. DoDo-Code is designed to fill this well-known gap in the literature.
>
> For larger values of $n$, the constant terms in the redundancy of combinatorial codes become negligible, and the existing combinatorial approaches are good enough for code rate. Therefore, DoDo-Code is not intended to be a universal solution, but rather a superior and "optimal" one for the specific, practical, and challenging domain of short-length codes.
>
> -----
> ## On the "Minimal Redundancy" Claim
>
> We want to clarify that the claim of achieving "minimal redundancy" is presented as a strong empirical observation rather than a formal, theoretical proof, as stated in the paper. This distinction was made explicitly in the paper, where we state the "assertion lacks theoretical proof" in Line 294. To reflect this, we used cautious phrasing, as seen in Line 284: “Section 4.2.2 DoDo-Code **may** represent the minimal redundancy achievable”.
>
> Even though, the evidence for this claim is compelling. As shown in Figure 5, the code rate of DoDo-Code almost perfectly aligns with the curve corresponding to a redundancy of $\log n+\log\log n+\log 3$ bits. This value can be reformulated as $\log 3n+\log\log n$, which is close to the theoretical lower bound of $\log 3n$ bits required to correct a single substitution error in a 4-ary alphabet. (The lower bound of $\log 3n$ is analyzed in Line 299.)
>
> It is obvious that there still be a remaining gap between the empirical redundancy of $\log 3n+\log\log n$ and the theoretical lower bound of $\log 3n$. Honestly, determining whether this $\log\log n$ term can be eliminated for 4-ary code is a significant theoretical open question beyond the scope of this work. However, there is evidence to suggest this term may be necessary. For instance, most of the combinatorial codes pursue an order-optimal redundancy of the form $\log n + O(\log\log n)$. The presence of a $\log\log n$ term in other solutions suggests it may be a characteristic feature. (Note that $\log 3n= \log 3 + \log n$ and $\log 3$ goes into $O(\log\log n)$)
>
> -----
> **Response to Questions**
>
> + At what code length n does the DoDo-Code become computationally impractical?
>
> The primary bottleneck is the codebook generation, which roughly scales with $O(4^n)$. Although this is a one-time, offline cost, the exponential scaling makes the greedy search computationally impractical for large values of $n$. Our experiments were conducted up to $n=11$ on a single CPU core, and we estimate that for $n$ significantly larger than $15$, the required resources would become prohibitive.
>
> + Does the performance advantage over existing methods persist at those scales?
>
> The trend of the performance in Figure 5 suggests that the advantage of the proposed work declines when $n$ grows. Actually, we do not expect the large margin performance advantage to persist for large $n$.
>
> As discussed above, the performance of existing combinatorial codes improve as $n$ increases, because the constant terms in their redundancy formulas become negligible. The order-optimal property of these codes makes them nearly optimal for large $n$. Therefore, for applications requiring large $n$, established combinatorial codes are good enough.
>
> Conversely, for applications that require small $n$, like segmented codes for multiple IDS error correction, DoDo-Code's advantage is clear. It provides superior performance at short lengths, a domain where other methods are poor in code rate.
>
> -----
> We hope these clarifications have adequately addressed the reviewer's concerns and have further highlighted the novelty of our work. We are confident that DoDo-Code offers a valuable contribution to the field and respectfully ask the reviewer to consider it for acceptance.

---

> > ### Comment · Reviewer_1PZb · 2025-08-03
> >
> > I thank the authors for the detailed response. Although I am not convinced that the space overhead is negligible, I will update my rating to 5.

---

> > > ### Author Response · Authors · 2025-08-04
> > >
> > > Thank you very much for your feedback and for the affirmation of our work.

---

### Official Review · Reviewer_Rv7D · 2025-07-02

**Clarity:** 3
**Significance:** 3
**Originality:** 3
**Rating:** 4
**Confidence:** 4

**Summary:**

This paper deals with coding for DNA storage. This problem is very popular in the research community as DNA based storage has a huge capacity. Sequencing operation is error prone and the errors which we can observe are not usual substitution errors but insertions and deletions. So, in order to deal with such errors new coding methods are required. Actually, such errors were already considered in the literature by Varshamov, Tenengoltz, Levenstein and others, these errors are a particular case of so-called asymmetric errors. But the problem is solved only for insertion or deletion of 1 symbol. The authors address the problem and propose new codes and embedding-based decoders that can deal with insertions, deletions and substitutions (IDS).

**Questions:**

- What is the memory complexity of K-d tree?
- Please mention the numbers of codewords in the constructed codes.
- What about long IDS correcting codes? Can the proposed approach be useful for this task?

**Ethical Concerns:**

["NO or VERY MINOR ethics concerns only"]

**Final Justification:**

I would like to thank the authors for clarification. I will keep my rating at 4.

**Limitations:**

The codes are short and contain small amount of codewords.

**Quality:**

3

**Strengths And Weaknesses:**

The paper is well-written, clear and proposes rather interesting approach.

The main weaknesses are as follows. The usual coding task is to construct exponential number of codewords, e.g. $2^{100}$. Here the number of codewords is small (such that we can perform brute force search).

The rate comparison is not clear for me. The VT codes are optimal codes that are guaranteed to correct one error. You perform comparison in different conditions (probabilistic ones). This should be mentioned explicitly.

---

> ### Author Rebuttal · Authors · 2025-07-28
>
> We would like to extend our sincerest gratitude to the reviewer for their time and insightful feedback on our manuscript. We are encouraged by the positive assessment that our paper is "well-written, clear and proposes rather interesting approach." We appreciate the opportunity to address the weaknesses and questions raised.
>
> Below, we address each of the reviewer's points in detail.
>
> -----
> ## On the Focus on Short-Length Codes and Codebook Size
>
> We thank the reviewer for their comment regarding the codebook size. This work deliberately targets a different, yet critical, area: designing high-rate, single-error-correcting codes for short lengths.
>
> This focus is motivated by two key observations from the existing literature and practical applications:
>
> **Suboptimality of Existing Codes at Short Lengths:** State-of-the-art combinatorial codes, while order-optimal for large n, exhibit poor code rates at short lengths. This is because the constant terms in their redundancy calculations become dominant when code length $n$ is small.
>
> **Practicality for Multiple Error Correction:** A common and practical strategy for correcting multiple IDS errors is to employ segmented error-correcting codes. In this paradigm, a long sequence is divided into shorter, disjoint segments, each designed to correct a single error. The overall efficiency and storage density of such a system are critically dependent on the code rate of the short codes used within each segment.
>
> Given these observations, high-rate, single-error-correcting codes for short lengths are critical for practical applications, **yet this area remains underdeveloped**. The proposed DoDo-Code directly addresses this gap by achieving a significantly higher code rate in the short-length regime. By improving the rate of these foundational short codes, this work provides a direct path to enhancing the performance of practical, multi-error-correcting systems.
>
> Regarding the specific number of codewords, we would like to direct the reviewer to Table 1 on Page 8 of the manuscript, which provides the precise cardinalities of the codebooks. For example, for a code length of $n=11$, our Deep Embedding-based Greedy Search (DEGS) finds a codebook with a maximum of $36,368$ codewords, a 16.8% improvement over the random search baseline. We will ensure this key result is referenced more clearly in the main text.
>
> -----
> ## On the Rate Comparison and Deterministic Error-Correction Guarantee
> We appreciate the reviewer raising this crucial point about the comparison conditions. We apologize if this was not sufficiently clear and welcome the chance to clarify a key aspect of our work: the DoDo-Code provides the same deterministic, worst-case guarantee for single-error correction as the combinatorial codes to which it is compared.
>
> The "probabilistic" nature of our approach lies in the search heuristic for finding a good codebook, not in the properties of the final code itself. The codebook construction method, as outlined in Section 2, ensures that the minimum Levenshtein distance d between any two codewords is at least 3. According to the principles of BDD, a minimum distance of d=3 is the requirement to unambiguously and deterministically correct any single IDS error. The comparison with VT codes is therefore conducted under the same conditions of a guaranteed single-error correction capability. It is worth noting that the VT code is optimal for correcting a single insertion or deletion over a binary alphabet. However, for the 4-ary alphabet with ID**S** errors, existing codes based on the VT construction are only order-optimal; no optimal code is known.
>
> The decoder, while leveraging the efficient K-d tree search in the embedding space, is also designed for high reliability. The tree search acts as a fast method to find the most likely candidate corrections. As we acknowledge that this search is based on an approximation, we employ a double-check mechanism by querying a small number of neighbors and verifying with the exact Levenshtein distance. As shown in Table 2, with $k\geq 4$, our decoder achieved a 100% success rate over $10^8$ trials, confirming its practical reliability for single-error correction.
>
> Although the proposed fast decoder (K-d tree search) makes a trade-off to achieve lower time complexity, the error-correction capability of the code itself remains deterministic. The code and BDD guarantee that a brute-force search over the codeword set will reliably find the correct sequence for any single IDS error.
>
> -----
> ## Responses to Specific Questions
> We thank the reviewer for the following specific questions:
>
> + What is the memory complexity of the K-d tree?
>
> The memory complexity to store the K-d tree is $O(m * |C(n)|)$, where $m$ is the dimension of the embedding vectors and $|C(n)|$ is the cardinality of the codebook. In our experiments, we used an embedding dimension of $m=64$, and the codebook sizes $|C(n)|$ are detailed in Table 1.
>
> + What about long IDS correcting codes? Can the proposed approach be useful for this task?
>
> This is a very relevant question. As we discuss in the Section 5, our current greedy search algorithm is computationally intensive and not directly suited for constructing codebooks for very long code lengths. For the long-code regime, existing combinatorial approaches are already order-optimal. Therefore, the primary contribution of our paper is for the short-length regime where our method demonstrates a significant rate improvement over the state-of-the-art. We believe, however, that the learned embedding space itself could be a valuable tool for future mathematical analysis to uncover principles that might lead to the invention of new, scalable codes.
>
> -----
> **Summary of Planned Revisions**
>
> Based on the reviewer’s valuable feedback, we will revise the manuscript to improve its clarity:
> + In the Introduction, we will add a sentence to more explicitly frame our work's focus on high-rate short codes and their direct relevance to practical segmented error-correction schemes.
> + In Section 4.2, we will add a clarification to state that the rate comparison is performed between codes offering the same deterministic single-error correction guarantee.
> + In Section 4.5, we will add the memory complexity of the K-d tree.
>
> -----
> Once again, we thank you for your thorough review and constructive suggestions. We are confident that these clarifications will strengthen our paper and better highlight its contributions.

---

> > ### Comment · Reviewer_Rv7D · 2025-08-05
> >
> > I would like to thank the authors for clarification. I will keep my rating at 4 as it is already rather high.

---

### Official Review · Reviewer_oBLi · 2025-07-03

**Clarity:** 3
**Significance:** 3
**Originality:** 3
**Rating:** 4
**Confidence:** 3

**Summary:**

This paper introduces DoDo-Code, a novel method introduced for designing high-code-rate single-IDS-correcting codewords through using deep Levenshtein distance embedding. The method leverages a Siamese neural network to map sequences (codewords) into an embedding space that preserves Levenshtein distances, then uses this space as a proxy for complex distance computations in both codeword search and error correction. The approach targets short-length codewords where existing combinatorial methods have poor code rates.
The authors make three key contributions: (1) a deep embedding method that approximates Levenshtein distances using squared Euclidean distance in embedding space, (2) a greedy codeword search algorithm that selects sequences (codewords) with lowest probability density function values to maximize code rate, and (3) a fast error correction method using K-d tree search in embedding space. Experiments on 4-ary codes demonstrate superior code rates compared to state-of-the-art combinatorial methods for short codewords.

**Questions:**

1. The paper lacks clarity about the exact filtering criterion in the greedy search algorithm. In Section 3.2, you describe "filtering out the neighboring sequences" without specifying the distance threshold. However, examination of the code reveals that all sequences with Levenshtein distance ≤2 from each selected codeword are removed, not just distance-1 neighbors as the term "neighboring sequences" typically implies. This critical detail should be explicitly stated in the paper for algorithmic clarity.
2. A pseudocode of the embedding-based greedy search algorithm may be included in the paper for better understanding.
3. Table 1 demonstrates that the revised loss function yields larger codebooks compared to the original loss when used with embedding-based greedy search. However, the paper lacks a clear explanation connecting the loss function design to this improved performance. Could the authors explain how the revised loss function's approach affects the embedding space structure and subsequent PDF-based codeword selection, ultimately leading to the larger codebooks observed
4. Could the authors provide more intuition behind the use of low probability density regions in the embedding space for codeword selection? A brief explanation how low-density embeddings correspond to fewer Levenshtein neighbors would make the strategy easier to understand.

**Ethical Concerns:**

["NO or VERY MINOR ethics concerns only"]

**Final Justification:**

This paper is well written and deserves to be presented at NeurIPS.

**Limitations:**

Yes, the authors do address the limitations.

**Paper Formatting Concerns:**

Nothing.

**Quality:**

2

**Strengths And Weaknesses:**

The paper presents a creative approach using deep learning by mapping sequences to embedding vectors and leveraging this embedding space as a proxy for the complex Levenshtein domain. A novel codeword searching algorithm was introduced in the embedding space to maximize the codebook, thereby achieving high-code-rate. Additionally, a decoding algorithm was integrated into the Euclidean embedding space using a K-d tree rather than operating in the Levenshtein domain, thereby reducing computational complexity.

Weaknesses: Some sections (e.g. 3.2) of the paper rely on concepts that are explained somewhat briefly. Providing more illustrative explanations or examples could improve clarity and make the overall approach easier to follow.

---

> ### Author Rebuttal · Authors · 2025-07-27
>
> We sincerely thank the reviewer for their thoughtful feedback and for recognizing the novelty and significance of our work. The reviewer’s comments are insightful and have helped us identify key areas where the manuscript's clarity can be improved. We are confident that by incorporating the suggested changes, the paper will be much stronger and more accessible.
>
> We will address each of the reviewer's points below.
>
> -----
> ## On the Filtering Criterion in the Greedy Search
>
> The reviewer points out that the paper lacks precision in describing the filtering step of the greedy search algorithm and that this detail is critical for reproducibility. We apologize for this oversight.
>
> To construct a code that can correct a single IDS error using a Bounded Distance Decoder (BDD), the minimum Levenshtein distance between any two codewords must be at least 3 ($d_{min} \geq 3$). This ensures that the decoding balls of radius $r = \lfloor((3-1)/2)\rfloor = 1$ are disjoint. To enforce this $d_{min} \geq 3$ property, the greedy algorithm selects a codeword and then filters out all sequences from the candidate set that have a Levenshtein distance of 1 or 2 from the selected codeword.
>
> In the revised manuscript, we will explicitly state this in Section 3.2. We will change the phrase "filtering out the neighboring sequences" to "filtering out the Levenshtein ball of radius 2 (i.e., all sequences $s’$ such that $\Delta_{L}(s_{codeword}, s') \leq 2$)" to remove any ambiguity.
>
> -----
> ## Request for Pseudocode for Section 3.2
> The reviewer suggests that pseudocode would improve the understanding of the greedy search algorithm. We completely agree with this.
>
> We will add a formal pseudocode block for the "Deep Embedding-Based Greedy Search" algorithm in the revised paper as follows.
>
> > #### **Algorithm: Deep Embedding-Based Greedy Search**
> >
> > **Input:**
> > * Codeword length $n$.
> > * The 4-ary alphabet $\Sigma_4$.
> > * A pre-trained normalized deep embedding model $f(·)$.
> >
> > **Output:**
> > * A codebook $C(n)$ where the minimum Levenshtein distance between any two codewords is at least $3$.
> >
> > **Procedure:**
> >
> > 1.  **Initialization**:
> >      * Create the candidate set $A=A(n)$ containing all $4^n$ possible sequences of length $n$.
> >      * Initialize an empty codebook $C=\Phi$.
> >
> > 2.  **Embedding and Distribution Fitting**:
> >      * Compute the embedding vectors for the candidate set: $U =\lbrace f(s) | s \in A(n) \rbrace$.
> >      * Estimate the covariance matrix $\hat{\Sigma}$ of the embedding vectors $U$ to model their distribution.
> >
> > 3.  **Iterative Greedy Selection**:
> >      * **While** the candidate set $A$ is not empty:
> >           * Select the sequence $s$ from $A$ whose embedding $f(s)$ corresponds to the lowest PDF value. This is achieved by finding the $s$ that maximizes the term $f(s)^{T} \hat{\Sigma}^{-1} f(s)$.
> >           * Add the selected codeword $s$ to the codebook: $C = C \cup \lbrace s \rbrace$.
> >           * Identify the Levenshtein ball of radius 2 around $s$, which is the set $B(s, 2) = \lbrace s' \in A | \Delta_{L}(s, s') \leq 2 \rbrace$.
> >           * Remove all sequences in $B(s, 2)$ from the candidate set $A$ to ensure the minimum distance requirement is met: $A = A \backslash B(s, 2)$.
> >
> > 4.  **Return**:
> >      * Return the final codebook $C(n)=C$.
>
> -----
> ## Explaining the Superiority of the Revised Loss Function
>
> The reviewer asks for a clearer explanation of why the revised loss function (Eq. 6) leads to larger codebooks compared to the original PNLL loss (Eq. 5), as shown in Table 1. This is an excellent question that gets to the heart of our model design.
>
> As clarified previously, the code's construction is dependent on local structures within the Levenshtein distance domain, namely sequences at a distance of 1 or 2. Consequently, from the perspective of the greedy search algorithm, there is no functional difference between using the complete Levenshtein distance and a truncated version.
>
> However, this distinction is raised for training the embedding model, $f(·)$. Relaxing the optimization objective to revised PNLL loss (Eq. 6) is an easier learning task for the model than approximating the global Levenshtein distance metric precisely. This allows the model to focus its capacity on the crucial local structure, providing a more accurate approximation for distances of 1 and 2.
>
> To be precise, the original PNLL loss (Eq. 5) aims for a good global approximation. It tries to be accurate across all Levenshtein distances, treating an error in estimating $d=10$ similarly to an error in estimating $d=1$. Our revised PNLL loss (Eq. 6) is specifically tailored for the BDD task. It focuses on creating a very clear margin in the embedding space around the critical decision boundary for single-error correction. It heavily penalizes two specific failures: (1) misidentifying a $d=1$ neighbor and (2) failing to push $d≥2$ sequences far enough away (i.e., predicting $\hat{d} < 2$ when $d≥2$).
>
> The revised loss function compels the model to learn a more structured representation by focusing on the decision margin. This enhances the accuracy of the subsequent PDF-based density estimation, which in turn enables the algorithm to pack more codewords into the final codebook $C(n)$. While our method using the original PNLL loss (Eq. 5) already produces a code rate approaching the theoretical optimum, the revised PNLL loss (Eq. 6) improves this rate further. As demonstrated in Table 1, this additional improvement is consistent, though marginal.
>
> We will add this detailed explanation to the end of Section 3.2 in the revised manuscript to connect the loss function's design directly to the empirical results in Table 1.
>
> -----
> ## Intuition for Low-Density Codeword Selection
>
> The reviewer asks for more intuition on why selecting codewords from low-density regions of the embedding space is an effective strategy.
>
> It is important to first note a key difference from conventional error-correcting codes. In codes based on the Hamming distance, for instance, the Hamming balls are the same size for every sequence. In that context, the choice of codeword does not need to account for ball size. However, for IDS error correction, Levenshtein balls are not uniform; their size depends on the corresponding center sequences.
>
> In greedy codebook construction, the strategy is to always select the codeword that removes the minimum number of other candidates. For our application, this translates to choosing sequences that have the smallest Levenshtein balls of radius 2, as this maximizes the candidate pool for future iterations.
>
> Unfortunately, characterizing the size of Levenshtein balls is a known hard problem in combinatorics, and no efficient algorithm exists to do this for every sequence.
>
> Our method uses the embedding space as a proxy to estimate this property. The Siamese network learns to map sequences with a small Levenshtein distance to vectors with a small Euclidean distance. It is therefore reasonable to assume that a sequence with many Levenshtein neighbors (a "dense" point in the Levenshtein domain) will be mapped to a vector that is surrounded by many other vectors (a dense region in the embedding space).
>
> The PDF of the fitted multivariate normal distribution is our formal measure of this density. Our core intuition is that these low-density vectors correspond to sequences that have fewer Levenshtein neighbors. By selecting these sequences first, the greedy search makes the efficient choice at each step, leaving maximal room for future codewords and thus maximizing the final codebook size. Figure 4a visually supports this, showing our method preferentially picks codewords from the distribution's periphery.
>
> We will integrate this step-by-step intuition into Section 3.2 to make the motivation behind our greedy search strategy clearer.
>
> ------
> **We believe these revisions will fully address the reviewer's concerns regarding clarity. We are grateful for the opportunity to improve our manuscript and are confident that the revised version will meet the high standards of NeurIPS.**

---

> > ### Comment · Reviewer_oBLi · 2025-08-07
> >
> > The authors clearly and thoroughly addressed all of my points, and I’m satisfied with their responses. I’m going to increase the score to 4.

---

### Official Review · Reviewer_hcVn · 2025-07-04

**Clarity:** 3
**Significance:** 2
**Originality:** 3
**Rating:** 4
**Confidence:** 4

**Summary:**

This paper introduces DoDo-Code, a deep learning–based method for constructing error-correcting codes over a 4-ary alphabet that are resilient to single insertion, deletion, or substitution errors. The core idea is to embed codewords into a continuous space where distances approximate the Levenshtein metric. This learned embedding enables efficient codeword construction through greedy search, and fast decoding using K-d trees. The motivation stems from the high computational complexity of directly computing Levenshtein distances in the discrete code space. By operating in a continuous proxy space learned by a neural encoder, the authors exploit the structured neighborhood geometry to select codewords more systematically, avoiding random or exhaustive search. For decoding, candidate codewords are first retrieved based on proximity in the embedding space, and then the exact Levenshtein distance is computed over this smaller subset to identify the closest match.

**Questions:**

No specific questions

**Ethical Concerns:**

["NO or VERY MINOR ethics concerns only"]

**Final Justification:**

Although the scope of work is of limited interest to NeurIPS community, it does tackle an interesting problem within the coding theory space and the community would find the work of value.

**Quality:**

3

**Strengths And Weaknesses:**

Strengths

1. Application is niche but novel. Successfully addresses the practical problem that existing combinatorial codes have poor rates at short lengths. Specifically, exploiting the multivariate embedding to replace random search over the codeword space with a more structured tree-based greedy search for error correction is very interesting.

2. Performance seems to be very good for short code lengths (N=10). While a code length of 10 is considered too small for general codes, within IDS codes it appears to be an acceptable choice for a small set of applications. Within this space, the practical performance (inference time for a given number of corrections) is impressive.

Weaknesses

1. Overly strong claims such as “...embedding vectors carry the properties of their original sequences from the Levenshtein distance domain...” are made without sufficient justification. Many such statements assume a high level of familiarity with the topic, making it difficult to fully appreciate the actual contributions.

2. The overall research theme seems more suited to an information theory or wireless communication venue than NeurIPS. The contributions from a deep learning perspective are few to none, with most innovations stemming from the clever application of deep learning to solve error correction problems. For instance, there is a large body of work on "neural channel codes," which similarly transform discrete codewords into continuous embeddings to improve error correction. Refer to [1], [2], [3].

Again, without detracting from the contributions of the paper, in my opinion, the work may be better appreciated by the coding theory community than the NeurIPS community, as the majority of the contributions lie in cleverly identifying the lack of structure needed for faster decoding of IDS codes in discrete space and bridging this gap by transforming the codes into continuous space using standard deep learning techniques.

1. ProductAE: Toward Training Larger Channel Codes based on Neural Product Codes, ICC 2022

2. DeepIC+: Learning Codes for Interference Channels, TWS 2023

3. Generalized Concatenated Polar Auto-Encoder: A Deep Learning Based Polar Coding Scheme, GLOBECOM 2024

---

> ### Author Rebuttal · Authors · 2025-07-25
>
> We thank Reviewer for the thoughtful and constructive review. We are encouraged that the reviewer found our approach to be novel, the application interesting, and the performance impressive for short code lengths.
>
> The reviewer’s main concerns appear to be
> + the paper's fit for the NeurIPS venue;
> + its relation to “neural channel codes”;
> + the clarity of certain claims.
>
> We address each of these points below and are confident that these clarifications will demonstrate our paper's suitability for NeurIPS.
>
> -----
> ## On the Contribution to Deep Learning and Fit for NeurIPS ##
> While we agree our work will be of interest to the coding theory community, we believe its core contributions also include a novel machine learning methodology for solving a difficult combinatorial search problem.
>
> + **A Novel ML-based Search Methodology**. The central contribution is not merely the application of a deep model, but the introduction of a new, learning-based heuristic for combinatorial search. The key insight is to learn an embedding space and use the probability density function (PDF) of the embedding vectors as a heuristic to guide a greedy search. This allows us to select candidate sequences from the periphery of the distribution, which we show leads to larger codebooks than random search. We believe this principle—using the learned density of an embedding space to guide a search—is a generalizable ML concept applicable to other combinatorial problems where "solution density" is a relevant but intractable property.
> + **Distinction from Standard ML Applications**. The custom loss function (Equation 6), is not a standard application of existing techniques. It is specifically tailored to the problem's unique requirement of ensuring a Levenshtein distance of at most 3. This demonstrates another way our approach is tailored to solve this specific scientific challenge.
>
> + **NeurIPS 2025 is an interdisciplinary conference**. We'd also like to highlight that the NeurIPS 2025 **Call For Papers**, explicitly welcomes submissions in areas of "**Applications**" and "**Machine learning for sciences**" and emphasizes that “The Thirty-Ninth Annual Conference on Neural Information Processing Systems (NeurIPS 2025) is **an interdisciplinary conference**…”. We believe our work, which introduces a new, learning-based heuristic for combinatorial search, fits squarely within this scope. While the application is in coding, the technique itself is a machine learning contribution, and we believe a top-tier, interdisciplinary venue like NeurIPS is the place to share this work with the broader community.
>
> -----
> ## On the Relation to “Neural Channel Codes”
> We thank the reviewer for pointing us to the literature on "neural channel codes." We have reviewed the suggested papers and agree they are important related work. However, they address a fundamentally different problem, and we will clarify this in our revision.
>
> + The neural channel codes (e.g., ProductAE, DeepIC+, GCC POLAR-AE) use autoencoders to learn an end-to-end communication system. The neural network itself is the encoder/decoder pair, trained to minimize the bit-error rate, effectively replacing classic codes.
> + DoDo-Code is a code construction algorithm. Our deep learning model is an offline tool used to search for a static, explicit codebook C(n) that satisfies a specific combinatorial property (d(C) ≥ 3). The final code is a simple lookup table, and the neural network is discarded after the search. Our method contributes to the design of codes, whereas neural channel codes contribute to their direct implementation as neural networks.
> + Another fundamental difference is the error channel our codes are designed for. Existing neural codes (e.g., ProductAE, DeepIC+, GCC POLAR-AE, ECCT) primarily target the Additive White Gaussian Noise (AWGN) channel and flip\substitution errors, a standard model for traditional communication systems. However, the AWGN model is not suitable for next-generation storage media, such as DNA-based storage or other biological sequence-based storage, where errors manifest as insertions, deletions, and substitutions. DoDo-Code is specifically built to handle these IDS errors, making the work distinct from and complementary to prior art in neural coding.
>
> We will add a paragraph to our related work section to discuss this important distinction, which will better contextualize our contribution.
>
> -----
> ## On Clarifying Claims and Improving Readability
> We appreciate the reviewer’s feedback on clarity. We agree that the statement “...embedding vectors carry the properties of their original sequences...” can be made more precise.
>
> To improve readability for a broader audience, we will revise this and similar sentences. For instance, we will change the above to:
>
> "The embedding aims to create a vector space where the squared Euclidean distance between vectors serves as a proxy for the Levenshtein distance between the original sequences. This allows us to leverage the geometric structure of the embedding space to reason about the complex combinatorial properties of the Levenshtein domain."
>
> We will perform a thorough pass of the manuscript to ensure our claims are precise and the work is accessible.
>
> -----
> We believe these revisions, combined with our clarifications on the contribution, will show that DoDo-Code is a novel paper well-suited for NeurIPS. **We hope the reviewer will consider our response favorably.**

---

> > ### Comment · Reviewer_hcVn · 2025-08-07
> >
> > I thank the authors for addressing my concerns. I updated my score accordingly.

---

### Note · Authors · 2025-08-12

We are grateful to the reviewers for the insightful discussion during the rebuttal period, which has significantly improved our manuscript. We are very encouraged to see the reviewers’ recognition of our work's strengths reflected in their reviews and the updated scores.

We will incorporate all suggested revisions into the final manuscript. Key changes will include:

+ discussing the relationship with the neural channel codes;
+ revising the Complexity section to include a more detailed space complexity analysis;
+ performing a thorough proofread of the manuscript to ensure precision and readability, include:
   + revising the statements on the relationship between the embedding and Levenshtein domains;
   + including pseudocode for key algorithms to improve clarity;
   + adding a detailed explanation of the revised loss function and the intuition for low-density codeword selection;
   + clarifying the motivation for the approach and the ideal application scope;
   + clarifying other statements as needed.

Once again, we are very grateful for the effort from the program committee throughout this valuable review process. We hope our work is considered to meet the high standards of NeurIPS.

---

### Decision · Program_Chairs · 2025-09-17

**Decision:**

Accept (poster)

**Comment:**

I wanted to thank authors and reviewers in engaging in a productive rebuttal discussion. All reviewers agreed that the paper presents acceptably novel and rigorous results, which also correlates with my own reading. I recommend acceptance.